# Aerobic Vaginitis—Underestimated Risk Factor for Cervical Intraepithelial Neoplasia

**DOI:** 10.3390/diagnostics11010097

**Published:** 2021-01-09

**Authors:** Olga Plisko, Jana Zodzika, Irina Jermakova, Kristine Pcolkina, Amanda Prusakevica, Inta Liepniece-Karele, Gilbert G. G. Donders, Dace Rezeberga

**Affiliations:** 1Department of Obstetrics and Gynecology, Riga Stradins University, LV-1007 Riga, Latvia; jana.zodzika@rsu.lv (J.Z.); opals19@inbox.lv (I.J.); dace.rezeberga@rsu.lv (D.R.); 2Gynecological Clinic, Riga East Clinical University Hospital, LV-1038 Riga, Latvia; kristine.pcolkina@inbox.lv; 3Faculty of Residency, Riga Stradins University, LV-1007 Riga, Latvia; aprusakevica@gmail.com; 4Pathology Center, Riga East Clinical University Hospital, LV-1038 Riga, Latvia; intaliepniecekarele@inbox.lv; 5Femicare Clinical Research for Women, 3300 Tienen, Belgium; gilbert.donders@gmail.com; 6Department of Obstetrics and Gynecology, University of Antwerp, 2550 Antwerp-Edegem, Belgium; 7Department of Obstetrics and Gynecology, Regional Hospital, 3300 Tienen, Belgium

**Keywords:** aerobic vaginitis, cervical intraepithelial neoplasia, abnormal vaginal microbiota, bacterial vaginosis, high grade cervical lesions

## Abstract

The aim of this study is to analyse the association between vaginal microbiota and the histological finding of CIN. From July 2016 until June 2017, we included 110 consecutive patients with abnormal cervical cytology results referred for colposcopy to Riga East Clinical University Hospital Outpatient department in the study group. 118 women without cervical pathology were chosen as controls. Certified colposcopists performed interviews, gynaecological examinations and colposcopies for all participants. Material from the upper vaginal fornix was taken for pH measurement and wet-mount microscopy. Cervical biopsy samples were taken from all subjects in the study group and in case of a visual suspicion for CIN in the control group. Cervical pathology was more often associated with smoking (34.6% vs. 11.0%, *p* < 0.0001), low education level (47.2% vs. 25.5%, *p* = 0.001), increased vaginal pH (48.2% vs. 25.4%, *p* < 0.0001), abnormal vaginal microbiota (50% vs. 31.4%, *p* = 0.004) and moderate to severe aerobic vaginitis (msAV) (13.6% vs. 5.9%, *p* = 0.049) compared to controls. The most important independent risk factors associated with CIN2+ were smoking (OR 3.04 (95% CI 1.37–6.76), *p* = 0.006) and msAV (OR 3.18 (95% CI 1.13–8.93), *p* = 0.028). Bacterial vaginosis (BV) was found more often in CIN1 patients (8/31, 25.8%, *p* = 0.009) compared with healthy controls (8/118, 6.8%), or CIN2+ cases (8/79, 10.1%). In the current study msAV and smoking were the most significant factors in the development of CIN in HPV-infected women, especially high grade CIN. We suggest that AV changes are probably more important than the presence of BV in the pathogenesis of CIN and progression to cervix cancer and should not be ignored during the evaluation of the vaginal microbiota.

## 1. Introduction

Persistent human papillomavirus (HPV) infection is a necessary cause for the development of cervical precancerous lesions (dysplasia) and cervical cancer [1]. Most of the cervical HPV infections resolve spontaneously, but in a minority of women it persists and progresses to cervical dysplasia and cancer [2]. Known factors that are associated with higher persistence of HPV infection and progression to high grade cervix lesions and cancer are smoking, use of oral contraceptives, impaired immunity and the concomitant presence of sexually transmitted infections (STI) [3].

In the recent years there have been publications about the association between different vaginal microbiota changes and cervical intraepithelial neoplasia (CIN), suggesting the idea that an abnormal vaginal environment plays an important role in the development of CIN [4,5,6,7]. The relation between bacterial vaginosis (BV) and cervical neoplasia has been studied most frequently: 20 studies about the association between BV and CIN had been analysed in a systematic review, but only 10 showed a significantly higher CIN prevalence in BV-positive women [8]. Much less is known about a possible role of other abnormal vaginal microbiota types such as aerobic vaginitis (AV) and mixed AV-BV flora. Aerobic vaginitis was first described in 2002 by Donders et al. [9] and is characterised by abnormal vaginal microflora containing aerobic and enteric pathogens, variable levels of vaginal inflammation and immature epithelial cells [10]. Nevertheless, AV is still often unrecognised or ignored; that could possibly be an explanation for some of the conflicting results for abnormal vaginal microbiota and cervical disease [6].

The aim of this study is to analyse the association between vaginal microbiota, including AV, and the histological finding of CIN.

## 2. Materials and Methods

In the time from July 2016 until June 2017 112 consecutive patients with abnormal cervical cytology results who were referred for colposcopy to Riga East Clinical University Hospital Outpatient department were included in the study group. 120 women who came for a gynaecological examination for any other reason (e.g., prophylactic examination, follow up of any gynaecological disease etc.,) were chosen as controls. The exclusion criteria were as follows: age under 18 years old, pregnancy, refusal to participate in the study. For the control group there was an additional criterion that women with a personal history of cervical precancerous lesions or cervical cancer were excluded. All women signed an informed consent prior to inclusion. The study was conducted in accordance with the Declaration of Helsinki and approved by the Ethical Committee of Riga Stradins university (Ethical approval code 39/24 November 2011). All confidentiality principles and patients’ data protection rules were observed.

Interviews, gynaecological examinations and colposcopies were done by a certified colposcopist (JZ, IJ, KP). During the interviews, the doctors filled a custom-designed questionnaire consisting of close-ended questions about demographic aspects, education level, smoking habits and the type of contraception. Primary (7–15 years of age) and secondary (16–18 years of age) education levels were combined and defined as ‘low education level’, as compared to ‘high education level’ (at least bachelors’ degree).

Un-moistened speculums were used during gynaecological examinations. A scraping was done from the upper vaginal fornix for pH measurement and performing wet-mount microscopy. Vaginal pH was measured on the glass slide with the Machery Nagel pH strips with a pH range of 3.1–7 [11]. Vaginal pH ≥4.5 was considered abnormal. Specimens for wet-mount microscopy were spread onto a glass slide, air-dried and later rehydrated with a drop of saline [12]. Microscopic examinations included the evaluation of lactobacillary grades (LBG), the number of leucocytes (less than 10 per high power field (hpf); >10/hpf, but <10/epithelial cell; ≥10/epithelial cell), the proportion of toxic leucocytes (none or sporadic, ≤50% of leucocytes, >50% of leucocytes), the presence of ‘clue cells’, red blood cells and sperm cells, the proportion of parabasal cells (none or <1%, ≤10%, >10%) and background flora (unremarkable or cytolysis, small coliform bacilli, cocci or chains) [9]. LBG were divided according to proportion between lactobacillus and other bacteria (Donders’ modification of Schröder’s classification [13,14]): LBG I—dominant presence of lactobacillus morphotypes, no other bacteria; LBG IIa—lactobacilli dominance, but other bacteria present; LBG IIb—other microorganisms outnumbering lactobacilli; LBG III—no lactobacilli, other bacteria present. LBG III was further divided to three subgroups: BV, AV and mixed BV-AV flora. A predominant granular microflora with uncountable bacteria all over the slide and >20% of ‘clue cells’ was defined as full blown BV, while mixed areas with streaks of BV-like microflora or sporadic ‘clue cells’ combined with other types of microflora were classified as partial BV [15]. The severity of AV was assessed using the AV score, described by Donders [9]. The score parameters were the following: LBG, the number of leucocytes, the proportion of toxic leucocytes, background microflora and the proportion of parabasal epitheliocytes. A composite AV score <3 represented no AV, the score 3–4—light AV, 5–6—moderate AV and >6—severe AV. Normal vaginal microbiota was defined as LBG I and IIa, but the abnormal as LBG IIb and III. To evaluate the effect of a definite abnormal microbiota type on the development of CIN, we divided pathological microbiota as follows: ‘any AV flora’ included LBG III AV, mixed AV-BV and LBG IIb with signs of AV; and ‘any BV’ included LBG III BV, mixed AV-BV and LBG IIB with signs of BV (partial BV). We have also assessed the severity grades of AV as a risk factor for CIN. Moderate to severe AV (msAV) was defined as AV score of 5 or more.

All study participants underwent a colposcopy, performed according to the local and European colposcopy guidelines. In the control group, if colposcopy was satisfactory and precancerous changes were absent, this result was interpreted as ‘no CIN’. From control patients with a visual suspicion for CIN and from all subjects in the group referred because of abnormal cytology, at least two biopsies were taken for subsequent histological analysis in the Riga East Clinical University Hospital Pathology centre. The results were classified as negative, CIN1, CIN2, CIN3 and carcinoma. In case of intermediate result, it was classified according to the more severe diagnosis (e.g., CIN1/2 was classified as CIN2). If there was a disagreement, the second reading was performed. Comparisons were made between all these groups of severity.

Statistical analysis was performed with Microsoft Excel 2010 and IBM SPSS 20.0. The relations between variables were assessed using Pearson chi-square or Fisher’s exact test. A *p*-value <0.05 was considered statistically significant. The relations between CIN2+ and different risk factors were assessed using univariate and multivariate logistic regression. Variables that showed a significant association in the univariate analysis (*p* < 0.05) were included in the multiple logistic regression. The risk of CIN2+ development depending on various risk factors was calculated as odds ratios.

## 3. Results

Four patients, who were initially in the control group, were excluded from the study because of the unreadable microscopy slides (*n* = 1) or an incomplete questionnaire (*n* = 3). Two patients in the study group had the histological result of cervicitis and therefore during data analysis were added to the control group. As a result, we have analysed 110 CIN1+ and 118 patients without CIN. The age range of the participants was between 19 and 59 years old. In the study group, women with lower education were more likely to be smoking (23/52, 44.2%) than women with higher education (15/58, 25.9%, *p* = 0.0001). This difference was not seen in the control group.

In the study group there were 31 (28.2%) cases of CIN1, 57 (51.8%) cases of CIN2, 21 (19.1%) cases of CIN3 and 1 (0.9%) cervical cancer case. There were no cases of glandular pathology found in the histological specimens. There were no differences found between the study group and the control group in terms of age, marital status, number of deliveries and the use of contraception method (Table 1). Women with cervical dysplasia (CIN1+) were more likely to have a lower education level and to be smokers.

Women with cervical pathology had more often increased vaginal pH (53/110, 48.2%) than women without CIN (30/118, 25.4%, *p* < 0.0001). Abnormal vaginal microbiota was encountered more often in the CIN1+ patients (55/110, 50.0%) than in the control group (37/118, 31.4%, *p* = 0.004). Any AV-associated microbiota changes and the proportion of msAV were significantly more frequent in the CIN1+ group (Table 2).

Abnormal vaginal flora, ‘any AV’-associated microbiota changes and patients with msAV were found significantly more often in patients with CIN2+ compared than in patients with CIN1 (Table 2). Compared with healthy controls (8/118, 6.8%), or CIN2+ cases (8/79, 10.1%), BV was found more often in CIN1 patients (8/31, 25.8%, *p* = 0.009) (Table 2). BV was not more frequent in the CIN2+ subjects than in normal/CIN1 patients combined (8/79, 10.1% vs 16/149, 10.7%, *p* = 1).

High education was less frequently encountered in women with CIN2+ (OR 0.39 (95% CI 0.21–0.71), *p* = 0.002) and there were more smokers (OR 3.96 (95% CI 1.88–8.33), *p* < 0.0001) in this group compared to women without dysplastic cervical lesions. Women with CIN2+ had more often msAV (OR 3.12 (95% CI 1.18–8.22), *p* = 0.021) than controls. (Table 3).

Multivariate logistic regression included analysis of education level, pH, smoking and msAV. It singled out smoking (OR 3.04 (95% CI 1.37–6.76), *p* = 0.006) and msAV (OR 3.18 (95% CI 1.13–8.93), *p* = 0.028) as the most important independent risk factors associated with CIN2+ (Table 3).

## 4. Discussion

In the current study we have found a significant association between low education, smoking, increased vaginal pH, abnormal vaginal microbiota and histologically proven cervical precancerous lesions. Two of the former associations, low education and smoking, are universal and are invariably encountered in several published studies [16,17,18,19]. Also, the correlation with elevated vaginal pH and abnormal vaginal microbiota on wet mount microscopy we found significantly more often in women with cervical precancerous disease, has been noted in the former work [4,5,6,7,20]. Vaginal pH measurement and wet mount microscopy are rapid and inexpensive point-of-care tests to evaluate vaginal environment [9,21,22]. In modern times, molecular biology techniques, such as a polymerase chain reaction (PCR), may seem to be a more detailed method for the evaluation of the vaginal microbiota. However, studies addressing accuracy of these tests describe comparable sensitivity and specificity of the wet mount microscopy [23,24], this test being more readily available and manifold more affordable in clinical practice.

In this study elevated vaginal pH was associated with CIN2+. Increased vaginal pH is mostly associated with the vaginal microbiota changes and infections, including sexually transmitted infections, and therefore could also be associated with sexual activity. However, there are also many non-infectious conditions that could alter vaginal pH, like low oestrogen level, the use of different vaginal products, hygiene habits (like vaginal douching), recent sexual activity and the presence of sperm or blood. These factors were not assessed in our study, which is a disadvantage.

The development of CIN requires prior HPV infection. After infection, however, there are many factors which play a role in the acquisition and persistence of HPV as well as in their virulence leading to the subsequent development of cervical lesions. Smoking is found to be one of these risk factors [19,25], as cigarette smoking impairs both cellular and humoral immunity [25,26]. As follows, disrupted cellular response may result into an inadequate clearance of the HPV leading to a persistent infection and progression to more severe disease [25]. In our data we confirmed that smoking was related to a three-fold increased risk of having CIN2+, and was the most important independent risk factor for CIN in multivariate regression analysis.

In our study, education level was inversely related to the risk of CIN2+, confirming other authors findings [27]. Poorly educated women are more often smokers [28], which is another independent cervical cancer risk factor.

Almost half of the study patients did not use any contraception method, and only a quarter used a male condom—the only method providing (partial) protection against STI and HPV [29].

Previous studies have shown a link between an altered vaginal microbiome and pre-invasive cervical disease [30,31,32,33]. The role of BV as a risk factor for CIN has been debated for many years. Some studies observed a significantly higher incidence of CIN in women with BV, but in these studies a different influence on low-grade or high-grade lesions was either not found or not assessed [8,33,34]. In one study, BV was not associated with low-grade lesions [5]. Our data revealed a link between BV and CIN1, but not with CIN2+. Low grade lesions can be caused by both high- and low-risk HPV [35,36]. Furthermore, sexual behaviour, such as exposure to a new partner, is the strongest risk factor for the infection with any HPV [35]. These associations suggest that BV can be considered as an indirect marker of sexual transmission of HPV, rather than a promotor of progression to more severe lesions. Carriage of *Gardnerella vaginalis*, a main constituent of BV, is induced by both penetrative and non–penetrative (oro-genital, digito-genital, vagina-vaginal) sexual contact [37]. In line with this, in a recent meta-analysis, BV was found to be a significant risk factor for HPV infection (OR 2.62, 95% CI 1.84–3.73, *p* < 0.05), but barely for CIN (OR 1.56, 95% CI 1.21–2.00, *p* < 0.05) [33]. Unfortunately, as our study was not designed to assess participants’ sexual activity and habits, we cannot prove our assumption.

Interestingly, and less well-known, is a relation between another abnormal vaginal microbiota type, aerobic vaginitis, and cervical precancerous lesions. AV, but not BV, was found to be significantly more often associated with abnormal cytology on Pap smears compared to normal cervical cytology [6]. In our study, for the first time, we observed that AV-associated microbiota changes were also linked to biopsy proven pre-invasive lesions of the cervix. Furthermore, there was a positive relationship between the severity of AV and that of the cervical HPV induced lesions: moderate to severe AV was strongly linked to CIN2+, while BV was not. This association remains strong when studied with multivariable regression analysis which indicated that msAV is linked to CIN2+ (3-fold more), alongside with smoking and low education.

In terms of pathogenesis, we hypothesise that the inflammatory characteristics msAV and HPV-induced cervix dysplastic lesions have in common are crucial for progression of the lesions in the direction of invasive cancer. Indeed, one of the critical mechanisms of HPV carcinogenesis is inflammation [38], shown by increased inflammatory interleukins (IL) in subjects with progressive CIN [39,40]. Also, inflammation of the uterine cervix is known to cause genotoxic damage through oxidation processes and changes in the cervical microenvironment [7,41]. Identically, AV is characterised by a different degree of inflammation, with increased vaginal leucocytes, presence of immune active ‘toxic leucocytes’ and highly increased concentrations of IL-1-β and IL-6 [9]. In the present study, besides a link with msAV, we also observed a non-significant positive correlation between the degree of leucocytosis and the severity of CIN2+ patients.

Our findings confirm the importance of a more in-depth analysis of the vaginal microbiota, such as description of AV and level of inflammation in the development of high-grade HPV-induced cervical lesions. We also confirmed that the finding of AV changes is probably more important than the presence of BV and should not be ignored during the evaluation of the vaginal microbiota. Hence, we strongly recommend that future studies addressing vaginal microbiome and cervix lesions should not be restricted to BV alone, as this may lead to incorrect conclusions. Similar interpretation errors have led to improved insights in the pathogenesis of vaginitis-related complications during pregnancy [42] and prevalence studies in African countries [43].

Whether or not the progression of CIN can be prevented by treating abnormal vaginal microbiota is still an open question. A recent study showed that increased numbers of *Prevotella bivia* and *P. disiens* could help to predict CIN2+ in HR-HPV-positive women [44]. Interestingly, recently *P. bivia* was also found to be one of the most important representatives in the microbiome of AV, alongside *Streptococcus agalactiae* [45].

The major limitation of the current study was its case-control design, which limited our ability to study the progression of lesions over time. As discussed above, also the lack of the information on the sexual habits and risk factors for respective infections were not mapped in this study, precluding the explanation of some links between BV and HPV. The strength of the study was its solid and reproducible outcome, as all women had a colposcopy and the severity of their lesions was proven by histology on biopsies. The relatively low sensitivity (51–57%) and specificity (66–76%) [46,47] of PAP tests in a screening setting for cervical dysplasia, as used in most previous studies, may either lead to the underdiagnosis of high-grade disease, or, particularly in the presence of AV, to the overdiagnosis of the cytological smear abnormalities [6,48].

## 5. Conclusions

We present new data indicating that msAV and smoking were the most significant factors linked to the development of cervical intraepithelial lesions. AV and associated inflammation seem to play a significant role in the carcinogenesis process on the cervix and should always be taken into consideration when interpreting vaginal microbiota in women with pre-invasive cervical lesions. In former research, as well as according to our data, increased prevalence of BV is limited to women with low-grade lesions only and can therefore rather be seen as an indirect marker of sexual behaviour leading to HPV infection than to be directly involved into the pathogenesis of cervical cancer.

Further studies are needed to fully understand the relation of abnormal vaginal microbiome with the development of cervical cancer in HPV-positive women, as well as to find the correlation between abnormal vaginal microbiota and other related risk factors (e.g., HPV genotypes, genetic susceptibility, etc.,). It is also important to clarify, whether or not the treatment of abnormal vaginal microbiota could help to prevent the development of cervix cancer.

## Figures and Tables

**Table 1 diagnostics-11-00097-t001:** Characteristics of the study (CIN1+) and control group (no CIN), % (*n*).

	Study Group with CIN1+, *n* = 110	No-CIN Group (Control + No CIN), *n* = 118	*p* Value
Mean age	35.7 ± 8.8	34.9 ± 9.3	0.583
Education level			
-primary & secondary	47.2 (52)	25.5 (30)	0.001
-high	52.7 (58)	74.6 (88)	
Marital status			
-married	55.5 (61)	51.7 (61)	0.848
-living with partner	35.5 (39)	38.1 (45)	
-no regular partner	9.0 (10)	10.2 (12)	
Smoking			
-yes	34.6 (38)	11.0 (13)	<0.0001
-no	65.4 (72)	89.0 (105)	
Number of deliveries			
-none	29.1 (32)	40.7 (48)	0.337
−1 delivery	37.3 (41)	30.5 (36)	
-≥2 deliveries	33.6 (37)	28.8 (34)	
Contraception method			
-Oral contraceptives	10.9 (12)	11.9 (14)	0.059
-Progestin-only pill	0	0	
-Intrauterine hormonal system	1.8 (2)	9.3 (11)	
-Other hormonal contraception	2.7 (3)	0.9 (1)	
-Intrauterine device	7.3 (8)	2.5 (3)	
-Male condom	24.6 (27)	25.5 (30)	
-Spermicides	0	2.5 (3)	
-Sterilisation	3.6 (4)	2.5 (3)	
-None	49.1 (54)	44.9 (53)	

**Table 2 diagnostics-11-00097-t002:** pH and microscopy findings according severity of CIN, % (*n*).

	No CIN	CIN1	CIN1+	CIN2+	No CIN vs. CIN1+	CIN1 vs. CIN2+	No CIN vs. CIN2+
	*n* = 118	*n* = 31	*n* = 110	*n* = 79	*p* Value		
Vaginal pH							
pH < 4.5	74.6 (88)	58.1 (18)	51.8 (57)	49.4 (39)	<0.0001	0.416	<0.0001
pH ≥ 4.5	25.4 (30)	41.9 (13)	48.2 (53)	50.6 (40)			
Lactobacillary grades							
LBG I	38.9 (46)	25.8 (8)	23.6 (26)	22.8 (18)	0.034	0.055	0.051
LBG IIa	29.7 (35)	29.0 (9)	26.4 (29)	25.3 (20)			
LBG IIb	15.3 (18)	12.9 (4)	23.6 (26)	27.9 (22)			
LBG III	16.1 (19)	32.3 (10)	26.4 (29)	24.0 (19)			
LBP I + IIa vs. LBP IIb + III	31.4 (37)	45.2 (14)	50.0 (55)	51.9 (41)	0.004	0.013	0.004
Vaginal Leucocytosis							
<10/hpf	69.5 (82)	80.7 (25)	69.1 (76)	64.6 (51)	0.996	0.460	0.703
>10/hpf, <10/epithelial cell	18.6 (22)	16.1 (5)	19.1 (21)	20.3 (16)			
>10/epithelial cell	11.9 (14)	3.2 (1)	11.8 (13)	15.2 (12)			
Presence of bacterial vaginosis (BV)							
-no BV	93.2 (110)	74.2 (23)	85.5 (94)	89.9 (71)	0.056	0.009	0.399
-any BV	6.8 (8)	25.8 (8)	14.5 (16)	10.1 (8)			
Presence of aerobic vaginitis (AV)							
-no AV	83.1 (98)	80.7 (25)	68.2 (75)	63.3 (50)	0.009	0.007	0.002
-any AV	16.9 (20)	19.3 (6)	31.8 (35)	36.7 (29)			
-msAV vs. no AV	5.9 (7)	6.5 (2)	13.6 (15)	16.5 (13)	0.049	0.006	0.002

**Table 3 diagnostics-11-00097-t003:** Results of univariate and multivariate logistic regression for the risk factors of CIN2+.

	Univariate Logistic Regression			Multivariate Logistic Regression		
	Odds Ratio	95% CI	*p* Value	Odds Ratio	95% CI	*p* Value
Smoking	3.96	1.88–8.33	<0.0001	3.04	1.37–6.76	0.006
Higher education level	0.39	0.21–0.71	0.002	0.42	0.21–0.81	0.009
msAV	3.12	1.18–8.22	0.021	3.18	1.13–8.93	0.028

## Data Availability

All relevant data are included in the published article.

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
