# Peer review of "Aerobic Vaginitis—Underestimated Risk Factor for Cervical Intraepithelial Neoplasia"

_diagnostics, 2021, doi:10.3390/diagnostics11010097_

Round 1

Reviewer 1 Report

The title of the manuscript sounds like pointed remark revealing the essence of the findings indicating aerobic vaginitis (AV) as underestimated risk factor for cervical intraepithelial neoplasia (CIN). Despite some limitations related with case-control design or lack of information concerning sexual habits of patients involved into current study associations between vaginal microflora and the histological findings revealed by authors are important indicating significance of specific vaginal microflora and its links within the pathogenesis of CIN. The conclusions indicating  smoking and aerobic vaginitis as factors most linked with cervical intraepithelial lesions in HPV infected women are correct and should be not ignored by other researchers.

Line 252: instead of "used is most" it should be written "used in most"   

Reviewer 2 Report

The manuscript is associated with the study of risk factors associated with cervical intraepithelial neoplasia in women. Therefore the manuscript is important for the practical aspects for better understanding of the etiology of cervical cancer. The main stress and declared novelty by authors is determination of association between Aerobic Vaginitis and progression of cervix cancer. The authors declare that such finding is novel however, there are data presented by other authors about such relation (for instance, 10.1007/s10096-016-2584-1). The manuscript is written in a classical way and has all necessary parts. The methods chosen are classical diagnostic methods for study of vaginal pathology. The methods are well described and statistical analysis presented in a very good way. I would say that the main shortcoming in this study was absence of modern methods in bacterial identification, particularly metagenomic analysis as concrete pathogens associated with Aerobic Vaginitis and cervical intraepithelial neoplasia remains unclear. Nevertheless the data about cervical cancer etiology was studied and conclusion presented which is also important for the knowledge support in this area. Below there are some remarks or questions that authors should answer: 1. Does the Ethical approval for this study was issued in 2011? If yes, does it was valid for 2016-2017 years? 2. I would recommend to mention the age of the women participated in the study. 3. It seems that the study was performed few years ago therefore some new data about diagnostics and etiology can be included into the part of Discussion. I would recommend to include short data about the possibilities of Next Generetion Sequencing for possibility to test bacterial communities and data about search of more concrete pathogens in this case. For instance, there are data that bacteria genus Sneathia can be associated with cervical carcirogenesis (https://doi.org/10.1038/s41598-018-25879-7). 4. The hypothesis described in rows 190-192 sounds too superficially for this research manuscript. I would recommend to drop the sentence "We hypothesize..." as well as the next sentence (rows 191-192), however this sentence can be left and rephrased in case the reference source will be added. 5. text in rows 195-198 (As we assume...) better fits for the project data or popular manuscript rather to this scientific manuscript therefore I suggest to drop it from the text. 6. I would recommend to change word "microflora" to "microbiota" in all of the manuscript. Flora means the species of Plant Kingdom, however, Bacteria were separated from Plants classification in 20 century so they are not recognized as plants anymore. 7. In the first sentence of Conclusions you present new data...in "HPV infected women" however, I was not able to find such data in the section Methods; how did you determined that women were infected with HPV? Please check and describe it in Methods if such data are absent. 8. row 259. The sentence should have more concrete ending ("when interpreting vaginal microflora"..please add more appropriate end such as "using classical methods" or "at routine diagnostics" or any other appropriate ending should be at the end of this sentence, because you can interpret vaginal microbiota by different reasons, for instance, when exploring microbiome of healthy or deseased women. As you have performed clinical diagnostics of pathological changes at your clinic you will find more appropriate words for this case.
